# Hallmarks of Cancer Affected by the MIF Cytokine Family

**DOI:** 10.3390/cancers15020395

**Published:** 2023-01-06

**Authors:** Romina Mora Barthelmess, Benoit Stijlemans, Jo A. Van Ginderachter

**Affiliations:** 1Lab of Cellular and Molecular Immunology, Vrije Universiteit Brussel, 1050 Brussels, Belgium; 2Myeloid Cell Immunology Lab, VIB Center for Inflammation Research, 1050 Brussels, Belgium

**Keywords:** MIF, DDT, cytokines, cancer, immunotherapy, hallmarks of cancer

## Abstract

**Simple Summary:**

The aim of this review is to summarize the available information regarding the MIF family in cancer, comprising MIF and DDT. Both cytokines are highly expressed in cancer patients, and their functions are related to 9 out of 10 hallmarks of cancer, suggesting that this cytokine family may become an important target to improve existing cancer therapies.

**Abstract:**

New diagnostic methods and treatments have significantly decreased the mortality rates of cancer patients, but further improvements are warranted based on the identification of novel tumor-promoting molecules that can serve as therapeutic targets. The macrophage migration inhibitory factor (MIF) family of cytokines, comprising MIF and DDT (also known as MIF2), are overexpressed in almost all cancer types, and their high expressions are related to a worse prognosis for the patients. MIF is involved in 9 of the 10 hallmarks of cancer, and its inhibition by antibodies, nanobodies, or small synthetic molecules has shown promising results. Even though DDT is also proposed to be involved in several of the hallmarks of cancer, the available information about its pro-tumoral role and mechanism of action is more limited. Here, we provide an overview of the involvement of both MIF and DDT in cancer, and we propose that blocking both cytokines is needed to obtain the maximum anti-tumor response.

## 1. Introduction

Over the past years, cancer therapy has become more personalized, starting with targeted therapy in the 1980s, followed by immunotherapy in 2010 [1], and gene therapy in 2017. The discovery of novel molecules involved in cancer progression allows for further stratification of patients, and corresponding therapies against these targets are being tested in pre-clinical and clinical studies. 

One of the most prominent examples of immunotherapy, which provided a breakthrough in the field, are the blockers of the immune checkpoint molecules, programmed cell death protein 1 (PD-1) and cytotoxic T-lymphocyte-associated protein 4 (CTLA-4). Both proteins are expressed on T cells, and, in homeostasis, they prevent auto-immune disease. However, in cancer, they inhibit the ability of T cells to kill cancer cells [2]. Consequently, the PD-1-blocking monoclonal antibodies pembrolizumab and nivolumab showed promising anti-cancer effects with tolerable adverse effects in [3] and are currently being used to treat non-small cell lung cancer (NSCLC), melanoma, bladder and colorectal cancer, and Hodgkin lymphoma, among others [4,5]. In addition, anti-CTLA-4 antibodies have been approved, such as ipilimumab, for colorectal, esophageal, and hepatocellular carcinoma, NSCLC, and renal cell carcinoma [6].

Another category of therapeutic targets are soluble factors that contribute to cancer progression. Well established examples are vascular endothelial growth factor (VEGF) and angiopoietin-2 (Ang-2), both of which are key regulators of angiogenesis and, hence, important cancer therapeutic targets [7]. Dual-inhibition blocking antibodies have resulted in reduced tumor growth, increased survival, and diminished vessel density, accompanied by a reprogramming of tumor-associated macrophages towards an anti-tumoral phenotype, in models of glioma [8]. Emerging soluble factors that have gained attention over the past years as contributors to tumor progression are the members of the macrophage migration inhibitory factor (MIF) family of proteins. This cytokine family comprises only two proteins in mice and humans: MIF [9] and D-dopachrome tautomerase (DDT), or MIF2, which shares a high structural homology with MIF [10]. These cytokines can be secreted by both cancer cells and stromal cells and have been linked to different types of cancer, with their increased expressions often being associated with a worse outcome [11]. For example, 31 samples of normal cervical tissue and 83 samples from cervical cancer tissue were analyzed for DDT expression via immunohistochemistry, showing that the DDT levels were significantly higher in cancer tissues. Moreover, the overexpression of DDT correlated with lymph node metastasis [12]. Similarly, MIF was highly present in 30% of breast cancer samples, while this was true for only 5% of the normal breast tissue samples. When patients were divided into MIF-positive and -negative groups, the MIF-positive group had a worse disease-free survival compared with the MIF-negative group [13].

In this review, we will discuss the functions of the MIF family members in cancer, and we will elaborate on how these cytokines could be considered clinically important immunotherapeutic targets. 

## 2. General Characteristics of the MIF Family

### 2.1. MIF Family Members

MIF is a highly conserved protein with evolutionarily ancient homologs in plants, protozoans, nematodes, and invertebrates [14]. The gene coding for MIF is located on chromosome 22 (22q11.2) in the human genome, and ~80 kb away is the gene for the more recently identified homolog DDT. DDT shares a ~34% pairwise sequence identity with MIF and most, but not all, of the biological activities of MIF [15]. A homolog of DDT, i.e., DDT-L, has more recently been discovered, but little information regarding this homolog is available as of yet. The *Ddtl* gene is also present on chromosome 22 in close proximity to *Mif* and *Ddt*, and its gene sequence overlaps with approximately 80% of the sequence of *Ddt* [16]. At the protein level, DDT-L and DDT consist of a total of 118 and 134 amino acids, respectively, and share 100% identity from amino acids 1 to 95. DDT-L is presumed to harbor lyase enzymatic activity, but no crystal structure of the protein is available so far, nor is there any clarity about its function [17].

### 2.2. Polymorphisms and Isoforms

Interestingly, two polymorphic sites that modulate the expression of MIF have been reported in the promoter region. Indeed, the microsatellite CATT_5–8_ in position −794 and the single-nucleotide polymorphism (SNP) in the −173 G/C position are linked to higher MIF protein expression. These polymorphisms have been associated with a higher risk of developing autoimmune as well as oncological diseases [18,19,20,21], highlighting the important role played by MIF in these pathologies. In the case of DDT, no polymorphisms in the promoter region have been described yet. 

In addition to polymorphisms at the genetic level, distinct post-translational variants of MIF exist as well, called oxidized and reduced MIF (oxMIF and redMIF). These conformational isoforms of MIF, which are dependent on the redox state of the microenvironment, were discovered by Scheiflinger’s group [22]. Notably, the implications of MIF oxidation are still debated in the field. The presence of redMIF was mostly found in the plasma and tissues of healthy individuals, whereas oxMIF was mostly present in patients with highly inflammatory diseases, in tumor tissues and corresponding metastases [23]. One of the oxidative modifications of MIF was shown in vitro, whereby myeloperoxidase (MPO)-derived hypochlorous acid and hypothiocyanous acid could oxidize the N-terminal proline (Pro-1) of MIF and alter its biological activity. Indeed, this modification led to a complete loss of tautomerase activity without affecting its immunomodulatory functions (i.e., increasing CXCL-8/IL-8 production via peripheral blood mononuclear cells (PBMCs) and blocking neutrophil apoptosis), indicating that tautomerase activity is not essential for these biological functions [24]. In vivo, different immune cells such as macrophages and neutrophils can contribute to the generation of reactive oxygen species (ROS) and pro-inflammatory mediators, oxidizing conditions that may also occur within the tumor microenvironment, wherein MIF could be converted to oxMIF. Remarkably, it has been suggested that redMIF could represent the latent form of MIF, whereas oxMIF might be the biologically active isoform [22]. However, given that Pro-1 is essential for CD74 binding, it could be argued that oxMIF (or at least MIF oxidized at Pro-1) is potentially less potent in triggering CD74 signaling, thereby annihilating several of MIF’s functions. 

Though the molecular identity of the putatively oxidized epitope remains to be demonstrated, certainly in vivo, it has been suggested, using the oxMIF-specific monoclonal antibody (imalumab), that this epitope could reside in proximity to the CALC motif. Interestingly, distal to the CALC motif is a third Cys (Cys-80), which operates as a switch, responsible for the shift from the reduced to the oxidized form, so it was termed the “switch cysteine”. In addition, the redox-sensitive amino acids within MIF (e.g., Cys-80 and lysine (Lys)-66) were identified as latent sites of this functional control [25].

Given that the levels of oxMIF within total MIF were significantly increased in the plasma of patients with different inflammatory diseases (sepsis, psoriasis, asthma, ulcerative colitis, Crohn’s disease, and systemic lupus erythematosus) versus control groups [23], oxMIF has been proposed as the “bad MIF”, which is, however, speculative. Additionally, since oxMIF was also specifically found to be expressed—using immunohistochemistry methods, which prevented an experimentally-induced conversion of MIF to oxMIF (e.g., by fixatives or oxidative agents)—in primary tumors (e.g., colorectal, lung, ovarian, and pancreatic) but not in adjacent non-tumoral tissues [26], it is considered a potential marker and target for cancer therapy [26,27]. However, the exact contribution of oxMIF in cancer, if any, as well as in inflammation, remains to be determined. 

### 2.3. Structure and Secretion

MIF and DDT have similar active homotrimeric structures (Figure 1) composed of monomers with very similar molecular weights: 12.5 kDa for MIF and 12,7 kDa for DDT [6]. Both trimers are expressed by a wide variety of cell types, including immune (T cells, monocytes, macrophages, dendritic cells (DCs), B cells, neutrophils, eosinophils, and basophils) and non-immune cells (epithelial and endothelial cells) [28]. An interesting characteristic that distinguishes MIF and DDT from other cytokines is their stimulation-independent expression. Indeed, MIF and DDT are constitutively expressed and are readily stored as active proteins within vesicles or in the nucleus of the cell [29]. As we will discuss further in this review, MIF has nuclease activity (Section 4.3), which could explain its presence in the nucleus [30]. Upon encountering a trigger, such as an infection or another inflammatory stimulus, these ready-made vesicular proteins are rapidly released to promote the activation of the innate and adaptative immune response [31,32]. As a matter of fact, MIF is incapable of being secreted via the canonical pathway since it lacks an N-terminal signal sequence for translocation into the endoplasmic reticulum (ER)/Golgi. On the contrary, MIF can be secreted by the Golgi-associated protein p115 (a vesicle-docking protein), the ATP-binding cassette transporters subfamily 1 (ABC1), and via pyroptosis following NLRP3 inflammasome activation. The interaction of p115 with MIF was discovered by using binding and immunoprecipitation experiments. After their interaction was confirmed, a knockdown of p115 in the THP-1 monocytic cell line was performed, resulting in a reduction in MIF secretion after LPS stimulation. This relationship was more evident in differentiated THP-1 cells, in which the reduction in MIF secretion appeared to correlate with the level of p115 downregulation [33]. Flieger et al. used different ABC1 inhibitors to test whether MIF secretion could be regulated by this transporter family. Indeed, glyburide and probenecid significantly reduced MIF release [34]. Previously, it was reported that MIF was necessary for the activation of the NLRP3 inflammasome [35,36]. Interestingly, pyroptotic cell death of immortalized bone marrow macrophages (iBMMs) following the activation of the inflammasome complex increased the release of MIF and IL1-β in a dose-dependent manner. Dankers et al. demonstrated that by blocking the NLRP3 inflammasome with specific inhibitors, MIF secretion was reduced [37]. Additionally, MIF can also be secreted in exosomes, such that exosome-derived MIF was found to regulate immune functions, such as the inhibition of reactive oxygen species generation and apoptosis and the promotion of metastasis [33,38,39,40]. In the case of DDT, the lack of an internal or N-terminal signal peptide in the mRNA sequence suggests that this cytokine is also secreted via a non-conventional secretion pathway, but the exact mechanism is still not known [41,42].

### 2.4. MIF Receptors: CD74

MIF and DDT can exert their functions through interaction with different receptors. The main receptor for both is CD74, which exists in three forms with different functionalities: (i) intracellularly, as MHC-II chaperone (referred to as the Invariant chain (Ii)) [43], (ii) at the plasma membrane, as the MIF/DDT receptor, and (iii) soluble/secreted as a MIF/DDT blocking form [44,45]. CD74 was first discovered as the invariant chain of MHC-II, which prevents the binding of endogenous peptides to the MHC-II peptide-binding groove in the ER. However, an excess of the invariant chain/CD74 is expressed compared with MHC-II, which is directed to the plasma membrane to function as a MIF and DDT receptor [46]. The majority of the CD74 protein produced stays intracellularly, and it is only around 2–5% of the total CD74 that is present as the membrane-bound form [46]. MIF and DDT share a conserved region around the proline-1 of their sequences, which allows them to bind to CD74. The proline-1 (P1) is located in an enzymatic pocket, rendering tautomerase activity to MIF and DDT. The mutation of P1 into a glycine causes MIF to completely lose its enzymatic activity and also affects its binding to CD74. MIF and DDT bind to CD74 with different affinities (1.40 × 10^−9^ M vs. 5.42 × 10^−9^ M, respectively) and binding kinetics. DDT has an 11-fold higher dissociation rate from CD74 than MIF, but it also associates about 3-fold faster with CD74 [41,46]. These differences in binding kinetics may influence signal transduction and may explain the differences in the dose-response profiles between MIF and DDT. Since the CD74 receptor does not have intracellular phosphorylatable regions, which are required for signal transduction, it needs a co-receptor. CD44 is a transmembrane receptor, which forms a complex with CD74, allowing the activation of the extracellular signal-regulated kinase 1 (ERK1)/ERK2, which are members of the family of mitogen-activated protein kinases (MAPKs), and the AKT/PI3K pathways [47]. These signaling pathways lead to the expression of genes involved in survival, proliferation, angiogenesis, and inflammation in response to MIF or DDT binding. Finally, CD74 can also be present in a soluble form (sCD74), which exhibits specific biological activities distinct from those of the membrane-anchored form. sCD74 is shown to be associated with a wide spectrum of diseases, including cancer [45]. Indeed, high levels of inflammation must be regulated to regain homeostasis; hence, MIF-regulatory mechanisms have been unveiled. One of the ways to accomplish this is by cleaving the extracellular region of CD74, thereby blocking the biological activities of MIF and DDT and thus decreasing the pro-inflammatory immune response. Recently, it was shown that the shedding of CD74 could be mediated in response to IFN-γ and mediated via an adrenomedullin (ADAM10 and ADAM17) and cysteine protease-mediated lysosomal cleavage. In turn, sCD74 exerts anti-survival and pro-apoptotic effects on CD74-expressing cancer cells. Though sCD74 was only shown to interfere with MIF [45], this molecule most likely also affects DDT’s biological activities.

### 2.5. MIF Receptors: Chemokine Receptors

MIF can also interact with chemokine receptors such as CXCR2, CXCR4, and CXCR7, which are present in different types of cells (monocytes, neutrophils, DCs, B cells, and cancer cells, among others), thereby promoting their migration [48,49]. For many years, it was thought that CD74 was needed for the interaction of MIF with these receptors, but more and more evidence contradict this, depending on the cell type. For instance, the binding of MIF to CXCR4 and CXCR7 and the subsequent signaling pathway activation were reported in CD74-deficient human rhabdomyosarcoma cells [28]. In addition, a peptide designed to specifically interact with the CXCR4-binding site of MIF selectively blocked the MIF/CXCR4 axis without interfering with the MIF/CD74 axis [50]. 

In contrast with MIF, DDT cannot bind to CXCR2 or CXCR4, as it lacks the regions needed for this interaction [41], in particular, (i) the pseudo-ELR and N-loop-like motifs for the CXCR2 interaction [50] and (ii) the RLR sequence at positions 87 to 89 and the N-loop-like motif for interaction with CXCR4 [51,52]. The impact of lacking these regions was suggested in a sepsis model, using MIF-deficient and DDT-deficient mice. In the MIF-deficient group, an increased survival associated with a reduced number of small peritoneal macrophages (SPMs), which express CD74 and CXCR2, was observed. Conversely, the lack of DDT had no impact on survival or on the numbers of SPMs in the peritoneal cavity, showing that only MIF is responsible for recruiting CXCR2^+^ SPMs, thereby promoting inflammation during sepsis [53].

However, the interaction of DDT and CXCR7 and the subsequent activation of signaling pathways have recently been reported. Indeed, ELISA and co-immunoprecipitation assays showed that the interaction of DDT with CXCR7 could activate the PI3K-Akt pathway, promoting the survival, cell growth, and proliferation of lung epithelial cells. Notably, the anti-apoptotic effects and phosphorylation of AKT by DDT were still prevalent when a blocking anti-CD74 antibody was used in [54]. 

### 2.6. Intracellular MIF-Interacting Proteins

In addition to binding to specific receptors at the cell surface, MIF and DDT may also bind to intracellular proteins and regulate intracellular signaling pathways upon endocytosis. For instance, during in vitro conditions with excessive MIF, the binding of MIF to the intracellular protein JAB1 (c-Jun activation domain-binding protein 1/COP9 signalosome subunit 5 (JAB1/CSN5)) inhibited its downstream signaling pathways [31], represented in the activation of AP-1 and the degradation of p27Kip1 [55,56]. Furthermore, DDT was shown to bind to JAB/CSN5 and possibly exert a similar effect on cell cycling as MIF. Moreover, recent evidence indicates that MIF is centrally implicated in the activation of inflammasomes. These aspects will be further discussed in the next section.

## 3. MIF and DDT Expressions in Cancer

It has now become clear that the binding of MIF and DDT to their receptors and/or intracellular targets has a strong effect on cancer promotion. This is corroborated by the fact that MIF and DDT expression is highly increased in cancer patients compared with healthy individuals in multiple cancer types, as illustrated in Table 1. Interestingly, MIF and DDT often play similar roles in tumor progression, independent of the cancer type. These cytokines stimulate cellular proliferation, inhibit apoptosis, and enhance angiogenesis. The sum of these roles mostly leads to a worse prognosis and increases the likelihood of developing metastasis. Notably, MIF was shown to be involved in all the 12 cancer types mentioned in Table 1, while DDT could only be demonstrated to play a role in 7 of them. This is likely reflecting a lack of knowledge about DDT rather than suggesting a lower importance of DDT, knowing that DDT can perform almost the same functions as MIF, which theoretically gives it the same capacity to affect tumorigenesis. This lack of information on DDT translates into a barrier preventing an appropriate generation of a full MIF family blockade strategy.

## 4. Hallmarks of Cancer Linked to the MIF Family

The hallmarks of cancer were described more than a decade ago by Hanahan and Weinberg, wherein they listed 10 biological characteristics acquired by cancer cells, which are key during tumor initiation, formation, maintenance, and further spread [83]. Recently, Hanahan published an updated version of these core hallmarks, mentioning four new characteristics: “unlocking phenotypic plasticity”, “non-mutational epigenetic reprogramming”, “polymorphic microbiomes”, and “senescent cells” [84]. Since MIF and DDT exert pleiotropic functions, it is not surprising that these cytokines have an intricate relationship with almost all the hallmarks of cancer (Figure 2). 

### 4.1. Evading Growth Suppressors

The transcription factor p53 is an important tumor suppressor protein that prevents the uncontrolled growth and division of cells. Indeed, p53 plays a main role in growth arrest, DNA repair, and apoptosis, so its dysfunctionality may lead to cancer [85]. Under homeostatic conditions, the intracellular protein levels of p53 are low compared with its mRNA levels. The main reason for this is the action of the E3 ubiquitin ligase Mdm2 (Mouse double minute 2 homolog), which promotes p53 ubiquitination and further degradation [86].

In 2003, Fingerle et al. used MIF-deficient mice in the C57BL/6 background to find a direct link between MIF and p53. Specifically, the lack of MIF in fibroblasts caused an earlier growth arrest, in a p53-dependent fashion, compared with the wild-type fibroblasts. In addition, when the MIF-deficient mice were confronted with carcinogens, they showed smaller tumors than the wild-type mice [87]. The interaction between p53 and MIF was also observed in glioblastoma, wherein the brain tumor-initiating cells (BTICs) showed high levels of MIF compared with the non-BITCs. Moreover, MIF was shown to physically interact with p53, inhibiting its functions. When *Mif* was silenced in BTICs, there was a decrease in cellular proliferation and in tumor formation [88]. 

Now, it is known that MIF and DDT have the ability to antagonize the actions of p53 in two different ways. First, MIF can, upon physically binding to p53, reinforce the interaction of Mdm2 with p53 [89,90], thereby increasing its degradation. Second, MIF and DDT may also indirectly influence p53. Cyclooxygenase isoenzyme 2 (COX-2) expression is activated by p53, which, in return, inhibits p53 [91]. Xin et al. showed that MIF and DDT can regulate the expression of COX-2 in a concentration-dependent way and that both cytokines are needed for maximal activity. MIF- and DDT-dependent COX-2 transcription was reported to require two pathways: the JNK/c-Jun pathway and the β-catenin/TCF pathway [92]. 

Another major tumor suppressor protein is Rb, which is an important regulator of the cell cycle. It stops the passage from the G1 to S phase by physically interacting with the transcription factor E2F, which controls the expression of genes involved in replication and in the progression of the cell cycle [93]. A relationship between MIF/E2F/Rb has been described, whereby human colon cancer cells were transfected with either sense- or anti-sense MIF-encoding plasmids. In the group transfected with the anti-sense plasmids, tumor growth was impaired. Most importantly, the transcriptions of Rb and E2F were downregulated when MIF expression was reduced and, conversely, upregulated when the MIF transcription was increased [94]. 

### 4.2. Resisting Cell Death

The induction of apoptosis is a well-studied pathway in cell biology. Briefly, after an apoptotic stimulus occurs, the Bcl-2 pro-apoptotic protein family becomes activated, the BID protein becomes truncated, activating BAX (i.e., a monomeric protein in the cytosol) or BAK (i.e., an integral mitochondrial membrane protein). Subsequently, the activated BAX or BAK proteins undergo a conformational change and oligomerize, after which they form pores in the outer membrane of the mitochondria, from which Smac (second mitochondria-derived activator of caspase) and cytochrome C translocate into the cytosol [95]. In turn, this leads to the activation of the caspase cascade, the executioners of apoptosis [96,97]. Interestingly, MIF and DDT have been shown to delay apoptosis by affecting several proteins in this mitochondrial mechanism. 

Initially, it was observed that the activation of PI3K/AKT, which is partly responsible for MIF’s signaling downstream of CD74, downregulated the expression of the pro-apoptotic genes *BAD* and *BAX* [98]. In addition to this, MIF’s silencing in cancer cells caused an increase in the release of cytochrome C, the downregulations of the Bcl-2 and Bcl-xL pro-apoptotic proteins, and increases in the pro-apoptotic proteins BAD and BAX and the tumor suppressor protein p53, overall, promoting apoptosis. These effects were proposed to rely on MIF’s binding to CD74, followed by the activation of NF-κB to control mitochondrial dynamics and stability. This, in turn, promoted carcinogenesis via impairing apoptosis [99].

Moreover, as well as prolonging the survival of cancer cells, MIF and DDT can also affect the longevity of immune cells, which was shown in macrophages and neutrophils. In a study by Mitchell et al., the administration of an endotoxin to MIF knockout mice augmented macrophage apoptosis in contrast with the wild-type controls, a phenomenon that was due to the inhibition of p53 accumulation when MIF was present [100]. In an in vitro study by Bauman et al., recombinant MIF given to neutrophils caused a delay in the cleavage of BID in a dose-dependent way. Additionally, MIF interfered with the release of the pro-apoptotic factors cytochrome C and Smac from the mitochondria and with the activation of the pro-apoptotic caspase-3, resulting in a longer neutrophil survival [101]. However, in a more recent study by Schindler et al., the anti-apoptotic effect of MIF (as well as DDT) in neutrophils was shown to be indirect [102]. Indeed, MIF and oxMIF, as well as DDT, prolong neutrophil survival, but only in the presence of mononuclear cells [103]. Mechanistically, MIF stimulated the release of soluble neutrophil survival factors, such as CXCL8, from peripheral blood mononuclear cells (PBMCs) upon interacting with CXCR2. Subsequently, MIF and CXCL8 needed to collaborate to promote neutrophil survival, since each factor separately is unable to reach this effect [102]. In a tumor context, MIF may promote the migration of neutrophils into the tumor which, in turn, may promote cancer progression. Hence, the promotion of tumor-associated neutrophil survival is another way this cytokine family promotes cancer development.

### 4.3. Genome Instability and Mutation

There is a constant, highly strict regulation in place that ensures cells remain in homeostatic conditions; however, cancer cells have acquired mechanisms to deviate from this regulation. To this end, mutations in key genes involved in regulating cell proliferation, the cell cycle, and DNA repair, among others, are the most common. The ways MIF and DDT are involved in evoking genome instability and mutations are various.

As mentioned before, MIF and DDT prevent p53 and Rb from performing their normal roles, leading to continuous proliferation, the accumulation of mutations, and the avoidance of cell death. However, this process also needs to be regulated, as it can cause stress due to the constant DNA replication and may jeopardize cell survival [104]. Cancer cells overcome this stress through different mechanisms acting on the DNA replication machinery, including the use of the poly (ADP-ribose) polymerase (PARP) family, comprising 17 proteins. One of the most important functions of PARP is to mediate single-strand break (SSB) repair, mainly carried out by the PARP1 protein [105]. If an SSB is not fixed, it will further progress into a double-strand break (DSB), which is considered lethal for cells. Consequently, PARP inhibitors have become a promising therapy for different cancers [106]. Interestingly, PARP1 recruits MIF, promoting their colocalization in the replication fork [30]. This could be of importance, since MIF was recently positioned as a 3′flap nuclease in charge of regulating DNA replication, having a direct impact on tumor growth. Two overhang structures are produced during DNA replication, the 5′ and 3′flap, and both need to be fixed to allow proper cell proliferation [107]. Wang et al. [30] found that MIF was able to specifically recognize the Y-shaped double-strand DNA to cleave 3′ flaps, proofread the DNA, and promote its elongation. These functions of MIF were confirmed in a MIF knockout breast cancer cell line. The loss of MIF nuclease activity led to increased DNA mutations and an impaired replication speed and cell cycle. 

### 4.4. Inducing Angiogenesis

One of the most studied factors contributing to angiogenesis is VEGF, the expression of which has been correlated with MIF. Indeed, MIF and VEGF were highly present in hepatocellular carcinoma patients compared with controls, both in the serum and in the cancerous tissues, and their expressions were positively correlated [108]. This close relationship between the expressions of MIF and VEGF was also reported in other types of cancer, such as gliomas [109]. A possible explanation for this close relationship was found in breast cancer cell lines. In these cells, the downregulation of MIF had a negative impact on the expression of VEGF-C, along with causing a reduction in the activation of the ERK/MAPK signaling pathway, which is needed for VEGF-C expression [110]. Similar data were obtained in vivo, upon the intraperitoneal injection of wild-type (WT, parental) versus MIF knockdown ovarian carcinoma cells. In addition to a reduced tumor growth and proliferation of MIF knockdown cells, the expression levels of keratinocyte chemoattractant (KC) and VEGF were also significantly reduced in the ascites of this group of mice [111].

A more direct link between MIF and angiogenesis was demonstrated by the group of R. Bucala, who showed that the administration of an anti-MIF monoclonal antibody to B-lymphoma tumor-bearing mice significantly reduced the tumor vascularization compared with the group receiving an isotype control. Tumors with similar sizes from both groups were studied, whereby the untreated group were found to contain 17.6 ± 5.8 capillaries/200X field and the treated group only 5.0 ± 2.5 capillaries. Notably, blocking MIF not only reduced the amount but also the diameter of new blood vessels [112].

Even though less information is available on the relationship between DDT and VEGF, some indications hint to the same effects as MIF. In a lung carcinoma study, the individual and cooperative downregulations of MIF and DDT had a negative impact on the expression levels of VEGF and CXCL8 [57]. In a non-cancerous setting, the hearts of cardiomyocyte-specific DDT knockout mice showed a 40% reduction in VEGF-A levels compared with WT mice. VEGF-A is a key enhancer of angiogenesis in the heart, after a transverse aortic constriction. Accordingly, the conditional knockout mice showed a lower capillary density compared with the controls [113].

In addition to directly regulating VEGF and angiogenesis, MIF and DDT can also indirectly influence these parameters. Indeed, MIF and DDT regulate and are regulated by hypoxia-inducible factor (HIF), a main transcription factor that induces the expression of genes involved in angiogenesis, such as VEGF, during hypoxic conditions [114]. This relationship between HIF and the MIF family members is likely to be important in cancer, since tumor hypoxia mediates many pro-tumorigenic effects, such as tumor progression, metastasis, and resistance to therapy [115]. The MIF–DDT–HIF axis is explained by the presence of a hypoxia-response element (HRE), which is a docking site for the HIF family of transcription factors, in the promoter region of the MIF and DDT genes. Consequently, MIF expression is increased in hypoxic environments, such as those present within a tumor, depending on HIF-1α [116]. In a feed-forward loop, HIF-1α expression is also up-regulated by MIF, mainly through the activation of PI3K/AKT signaling [117] and by preventing its ubiquitination and further degradation [118]. Notably, the promoter region of the *Ddt* gene also contains three HRE regions, and it has been shown that HIF-1α, as well as HIF-2, are associated with the DDT promoter region under hypoxic conditions [77].

### 4.5. Activating Invasion and Metastasis

One of the features of aggressive cancers is their ability to detach from their initial niche, enter and exit the bloodstream, and adapt to a new environment to establish metastases. The high levels of MIF present in tumors, in addition to providing an advantage in terms of tumor growth, also grant the cancer cells a particular aggressiveness, making them more prone to migrate and metastasize. This pro-metastatic role has been reported in several types of cancer.

For instance, Yang et al. reported that MIF inhibited NR3C2, a tumor suppressor gene that encodes a mineralocorticoid receptor. With reduced levels of NR3C2, the growth, migration, and invasion of cancer cells were increased [119]. In addition, MIF was shown to correlate with the increased epithelial-to-mesenchymal transition (EMT) of the cancer cells [119]. EMT is the process by which cells lose epithelial characteristics while gaining mesenchymal features. This is needed during early development; however, cancer cells use this process to their advantage [120]. Yang et al. found that cells with high levels of MIF had a decreased expression of E-cadherin and an increased expression of N-cadherin, vimentin, and Zeb1 [119]. This switch in cadherin expression is the main feature of EMT, whereby E-cadherin preserves the integrity of the epithelial structure, while N-cadherin promotes migration [121]. Huang et al. found that downregulating MIF expression with short-hairpin RNA (shRNA) inhibited the EMT by promoting the opposite process, i.e., mesenchymal-to-epithelial transition (MET) [122]. In the case of salivary adenoid cystic carcinoma (SACC), the silencing of MIF expression significantly reduced the occurrence of EMT in a SACC cell line [123]. This was also observed in human glioma cells, in which the overexpression of MIF incremented the expression of mesenchymal markers. The increase in EMT was also tested in vivo, whereby the administration of recombinant human MIF caused an increment in tumor size and in EMT. Moreover, this effect was inhibited when the CXCR4-AKT pathway was blocked [124].

Currently, there is no clear relationship between DDT and EMT, but the effects of DDT in promoting invasion and metastasis have been reported. Using DDT shRNA in a pancreatic cancer cell line resulted in a reduced invasive capacity of these cells. When a double knockdown of MIF and DDT was performed, a further reduction in invasion was observed [78]. Similar results were found in human cervical cancer cells, with some level of migration and metastasis reduction upon DDT knockdown but stronger effects were achieved when MIF and DDT were knocked down together [12]. These data, again, suggest the capacity of MIF and DDT to compensate for each other’s absence. 

### 4.6. Deregulating Cellular Energetics

Under physiological conditions, cells use glycolysis to degrade glucose into pyruvate for the generation of ATP. However, cancer cells have a higher energy demand and therefore have modified their cellular energetics. They preferentially perform aerobic glycolysis as an alternative energy-generating metabolic pathway, whereby glucose is converted into lactate, even in the presence of oxygen and functional mitochondria. This phenomenon is referred to as the Warburg effect [125]. One of the main enzymes involved in the Warburg effect is lactate dehydrogenase A (LDHA), which is regulated by HIF-1α and is responsible for the conversion of pyruvate to lactate [126]. The secreted lactate can then be taken up by other cancer cells, but also stromal cells such as macrophages [127], as their main energy source and can be considered as a vehicle for transferring energy to different cancer subpopulations (residing in normoxic versus hypoxic regions within the tumor) [127].

Several studies have highlighted a relationship between MIF, cancer cells, and the Warburg effect, demonstrating a correlation between high MIF levels and dysregulated cellular energetics. Li et al. studied this relationship in a lung cancer context, showing a correlation between the levels of MIF and LDHA, with an increase or decrease in LDHA depending on the overexpression or knockdown of MIF, respectively [119]. A same type of relationship was found between MIF levels and the Warburg effect. It was further shown that MIF modulates the Warburg effect through HIF-1α. As explained before, a positive feedback loop between MIF/DDT and HIF-1α exists. Additionally, HIF-1α, as well as promoting angiogenic gene expression, is also implicated in the regulation of metabolic genes involved in cellular energetics and the Warburg effect, as reviewed by Courtnay et al. [128]. In line with these findings, Li et al. showed that overexpression of MIF increased HIF-1α and the Warburg effect in a lung cancer model. However, when HIF-1α expression was silenced, the Warburg effect was no longer upregulated [129].

Up to now, no relationship between DDT and the Warburg effect has been reported. However, because this cytokine also promotes the expression and stabilization of HIF-1α, it is quite possible that DDT also has a promoting effect on the Warburg effect. 

### 4.7. Sustaining Proliferative Signaling

Several publications have shown that the knockout or knockdown of MIF and/or DDT has a direct effect on cell proliferation. For instance, in renal cancer cell lines, the individual, as well as the combined knockdown of MIF and DDT, affected many cancer features, such as the invasion, migration, and proliferation rates. A proliferation assay showed that the DDT knockdown cancer cells had a 50% reduction in proliferation in monolayer conditions due to increased doubling times (control: 45 h, DDT knockdown: 52 h, MIF knockdown: 55 h, and MIF/DDT knockdown: 68 h), with the double knockout group being the one with the greatest increase [77]. Similarly, in a colorectal cancer cell line, the knockdown of MIF resulted in a significant reduction in proliferation. The authors discuss that the signaling pathway involved in this effect could partially be AKT/GSK-3β, as its phosphorylation was impaired when MIF was downregulated [122]. Moreover, it was previously shown that inhibition of GSK-3β, a serine/threonine kinase involved in different cellular processes, can reduce colorectal cancer cell proliferation by interfering with the expression of NF-κB [130].

Charan et al. studied the effect of reducing MIF production via TNBC cells. Transducing these cells with MIF shRNA caused a decrease in cancer cell proliferation compared with the control. In addition, the administration of a small synthetic MIF inhibitor, CPSI-1306, to cancer cells in vitro had a detrimental effect on their proliferation and survival. Also in vivo, mice that received this inhibitor were found to bear smaller tumors compared with the control, which was found to correlate with a reduced Ki67 expression in the cancer cells [61].

In 2019, Bucala’s group screened in silico 1.6 million compounds for their ability to target DDT, and identified 4-CPPC as the best candidate, showing a high inhibitory potency and specificity towards DDT. This small compound was tested in vitro and found to inhibit the DDT/CD74 signaling pathway [131]. Recently, a new synthetic compound highly selective for DDT was reported, i.e., Thieno[2,3-*d*]pyrimidine-2,4(1*H*,3*H*)-dione Derivative (or d5). This inhibitor significantly reduced the proliferation of non-small cell lung cancer cells in 2D and 3D cultures. The mechanism behind this effect relied on the known signaling pathways activated by DDT, such as the MAPK signaling pathway, which regulates the expression of genes involved in the cell cycle [60].

### 4.8. Tumor-Promoting Inflammation

Inflammation and cancer have an intricate relationship, but more information has been gathered in the last decade to better understand how inflammation impacts tumorigenesis [132]. The chronic exposure of cells to inflammation can lead to cellular transformation. In addition, inflammatory cells within the tumor microenvironment may contribute to the production of tumor-promoting molecules, such as growth factors, pro-angiogenic cytokines, and enzymes that can promote invasion (e.g., CSF-1, EGF, VEGF, and MMP-9) [133,134]. 

By themselves, MIF and DDT are pro-inflammatory cytokines. As a matter of fact, MIF is considered a pro-inflammatory regulator, which is often at the basis of initiating an inflammatory response. These cytokines have the ability to activate the innate and adaptive immune compartments and regulate, directly and indirectly, the secretion of other pro-inflammatory molecules such as IL-2, IL-6, IL-8, TNF, IFN-γ, and IL-1β [15]. Additionally, MIF overrides the anti-inflammatory effects of glucocorticoids by preventing their expression [28]. Hence, although MIF and DDT are needed to mount an efficient inflammatory response against infectious and non-infectious insults, their excessive production may be detrimental. Indeed, these cytokines are not only bad prognosticators for cancer, but also worsen autoimmune and inflammatory pathologies such as multiple sclerosis, burns, and arthritis [41,135,136,137].

### 4.9. Avoiding Immune Destruction

In addition to being an initiator of inflammation, MIF is also reported as a major immunosuppressive factor [103], although this may be restricted to certain contexts such as cancer. In cancer, the use of MIF knockdown or knockout cancer cell lines has taught us that, depending on the tumor model used, different immune cells could be affected by MIF-signaling. 

In a metastatic melanoma model, it was shown that blocking the interaction of MIF and CD74 with peptide-based therapy reduced the immunosuppression in different immune cells and decreased the number of lung metastatic foci in immunocompetent mice. Within the innate compartment, more M1 macrophages (macrophages with an anti-tumoral phenotype), as well as intratumorally activated DCs, were found in the lung metastases of the treated group compared with the control group. In the lymphoid compartment, more cytotoxic CD4^+^ T cells (13.35% vs. 6.86%), CD8^+^ T cells (17.6% vs. 6.11%), and NK cells (16.13% vs. 8.44%) were found in the treated group compared with the control group [138]. In a colon carcinoma model, MIF knockout mice showed smaller tumors than the WT group, along with significantly fewer Tregs within the tumors and spleens. Mechanistically, MIF was shown to upregulate IL-2 expression, which is mainly used by Tregs, resulting in their expansion [139]. These data point to the role of stromal-cell-derived MIF, but also cancer-cell-derived MIF can play a role. Indeed, a MIF knockout breast cancer cell line led to smaller tumors upon injection in mice compared with the control cell line. Within the tumor microenvironment, increases in intratumorally activated DCs and IFN-γ-producing CD4^+^ and CD8^+^ T cells were found in the group inoculated with the MIF-deficient cancer cells [62]. Along the same line, Tessaro et al. used a WT and an MIF-silenced sarcoma cell line for in vivo tumor growth, showing that the tumors from the MIF-silenced group were more prominently infiltrated by CD45^+^ hematopoietic cells, with a particular enrichment of monocytes and CD4^+^ T cells. At the transcriptional level, macrophages from the MIF-silenced tumors resembled the more pro-inflammatory, anti-tumorigenic MHC-II^high^ macrophages, with an increased expression of inflammatory and antigen presentation-linked genes. In addition, the CD4^+^ T cells expressed higher levels of IFN-γ and the Tbx21 gene, which encodes the transcription factor T-bet, which is characteristic of Th1 cells. Finally, although the contribution of CD8^+^ T cells to the tumor-immune infiltrate did not increase, these cells showed a more activated phenotype with upregulations of various activation markers [140]. 

Another cell type known to acquire a pro-tumorigenic phenotype within the TME is the neutrophil; MIF, through the binding of CXCR2, promotes their tumor infiltration. Moreover, it will also induce the production of CCL4 and MMP9 with this immune cell type, leading to the promotion of lymph angiogenesis and the remodeling of the tumor stroma [141]. Lathia’s group showed an immunosuppressive role of MIF via influencing MDSCs in glioblastoma. Two subsets of MDSCs expressing different MIF receptors were described: a monocytic MDSC, expressing high levels of CD74, with a large presence within the tumor microenvironment, and a granulocytic MDSC, expressing CXCR2, with a low presence in the tumor. The blocking of MIF with a small compound, ibudilast, allowed a reduction in MDSC generation and function and, at the same time, an increase in CD8^+^ T cell function [74].

Considering all this information, it is clear that MIF promotes tumor establishment and impairs the ability of the immune system to fight back. Most importantly, independent of the tumor models used, similar immunosuppressive functions related to MIF have been observed, suggesting that MIF could be a potential target for several cancer types. 

### 4.10. Unlocking Phenotypic Plasticity

As described by Hanahan [84], escaping from terminal differentiation is critical for cancer cells. There are three different ways to accomplish this: (i) dedifferentiation, which is the mechanism whereby already matured, fully differentiated cells return back to a progenitor state, (ii) blocked differentiation, implying that the differentiation process is inhibited, and the cells stay in a progenitor state, and (iii) transdifferentiation, whereby the cells differentiate into another lineage. This last type of differentiation has already been partially mentioned in Section 4.5, as EMT is a type of transdifferentiation that leads to invasion and metastasis [142].

In the case of glioblastoma, the mesenchymal transdifferentiation of glioblastoma multiforme (GBM) cells has been thoroughly studied. This transition has been associated with acquired resistance to therapy, causing a worse prognosis, and can be regulated in different ways [143]. NF-κB is a key regulator of this transition, and as mentioned above (Section 4.2 and Section 4.7), MIF is involved in the activation of this transcription factor. Consequently, inhibition of MIF could prevent the process of transdifferentiation. Accordingly, the combination of radiation and 4-IPP, a small compound capable of blocking MIF and DDT, prevented the increase in the mesenchymal markers TGM2 and NF-κB compared with when only radiation was used [11].

Another way to induce cancer cell transdifferentiation is through hypoxia. The group of Frank A.E. Kruyt showed the acquisition of mesenchymal markers by non-mesenchymal cancer cell lines upon exposure to hypoxia. In addition, the induction of HIF-1α expression stimulated mesenchymal transition, whereas the knockdown of HIF-1α by shRNA interfered with this phenomenon [144]. As previously stated, there is a direct correlation between MIF/DDT and HIF-1α (Section 4.4 and Section 4.6), with MIF secretion during hypoxic conditions also being reported in glioblastoma [124]. From these findings, we infer that MIF can act via two different pathways to promote cancer cell transdifferentiation. Even though there is no available information regarding DDT in this respect, we do not rule out its possible involvement, as DDT is also associated with NF-κB and HIF-1α functionality. 

### 4.11. Senescent Cells

The main definition of senescence is the arrest of cell division and growth. This directly impacts the energetics and metabolism of cells, causing changes in their phenotype, which is called the senescence-associated secretory phenotype (SASP). The major secreted molecules by SASP cells are chemokines, pro-inflammatory cytokines, extracellular matrix components, growth modulators, and matrix metalloproteinases [145]. At first sight, senescence seems to be a beneficial effect that protects against tumor growth, but some findings suggest that it can also promote tumor progression, making it a double-edged sword [146]. 

Saul et al. carried out an extensive bioinformatic analysis and developed a senescence gene expression panel that was used to identify senescent cells and their intracellular signaling pathways. SASP cells were found to highly express the MIF pathway members *CD74, CXCR4,* and *CD44.* Moreover, scRNA-seq data showed that hematopoietic and mesenchymal cells had *Mif* as a major SASP gene regulator, which increased during senescent cell burden and was reduced along with the clearance of senescent cells. In vivo data corroborated these results, as the expression of MIF was higher in bones from old mice, and the levels were reduced upon the genetic clearance of senescent cells. Since high levels of MIF expression have been linked to immune evasion, it could be argued that the augmented expression of MIF in senescent cells could affect their immune clearance, conferring on them some level of immune resistance [147]. 

## 5. MIF and DDT as Novel Targets in Cancer Therapy

Considering the involvement of MIF and DDT in multiple hallmarks of cancer, this cytokine family could be considered a promising therapeutic/pharmacological target in cancer treatment. To date, several strategies have been developed to block MIF and/or DDT signalings, such as small synthetic molecules, antibodies, and peptide-based receptor-blocking molecules. 

Until now, the tautomerase activity (which is shared by MIF and DDT) has most often been used as a target, with the identifications of specific inhibitors. As mentioned in Section 2, the proline-1 residue is not only critical for the enzymatic activity, but also for the binding of MIF/DDT to CD74, providing a rationale for targeting the active site as a MIF/DDT blocking strategy. Notably, no physiological substrate for MIF or DDT’s enzymatic activity has so far been found [148]. The small-molecule synthetic inhibitors that have been developed to block MIF’s tautomerase activity can be classified into different categories: (i) non-covalent inhibitors that bind to the tautomerase active site, (ii) covalent inhibitors that bind to the tautomerase active site, (iii) allosteric inhibitors that disrupt MIF’s trimeric structure or induce a conformational switch, and (iv) stabilizers of monomeric MIF [149]. 

Due to the fact that these small compounds are non-immunogenic, have a low manufacturing cost, exhibit efficient tissue penetration, and can be given orally, this pipeline of inhibitors has gained great interest. However, they show a different half-life compared with monoclonal antibodies due to differences in their elimination rates and metabolic conversions. For instance, in the case of EGFR-targeted cancer drugs, the small compounds erlotinib and gefitinib exhibit short half-lives of 36 h and 48 h, respectively, compared with the longer half-life of the monoclonal antibody cetuximab (3.1–7.8 days). This difference in half-life impacts the dosing strategy of each compound, urging for more frequent dosing in the case of small compounds [150]. Up to now, the effects of MIF/DDT small-molecule inhibitors have mainly been tested in vitro and in several murine preclinical tumor and inflammatory models. The most relevant preclinical finding, providing a basis for their clinical use, is the significant tumor reduction observed when they are administered to tumor-bearing mice. This is mainly due to a reduction in the proliferation, migration, invasion, and survival of the cancer cells [16,61,151]. Genetic studies have complemented the findings of these inhibitor studies. Indeed, upon assessing the frequency of tumor development in a skin-carcinogenesis model in WT, P1G MIF, and MIF knockout mice, the P1G-MIF group (being deficient in tautomerase activity and binding to CD74) showed an intermediate frequency of tumor development, in between the WT and the knockout group [152]. Because MIF is involved in several inflammation-related diseases, MIF inhibitors have gained great interest and, in some cases, have already been approved for treating conditions other than cancer. This is the case for ibudilast, an anti-inflammatory drug and allosteric inhibitor of MIF, which is already approved in Japan for the treatment of bronchial asthma and cerebrovascular disorder. Regarding cancer, the effect of this small compound was recently also studied in glioblastoma, the most aggressive type of malignant brain cancer [153]. The combination of ibudilast and temozolomide (the standard-of-care chemotherapy to treat GBM) caused cell cycle arrest and increased the apoptosis of patient-derived cell lines in vitro and prolonged the survival of a patient-derived xenograft model in vivo [154]. Currently, this combination is being tested in clinical trials for glioblastoma (NCT03782415). 

A downside of the inhibitor approach is that all compounds tested so far are specific for MIF (e.g., ISO-1) or exhibit only limited cross-reactivity with DDT (e.g., 4-IPP) [155]. However, more inhibitors that selectively target DDT are being discovered (e.g., 4-CPPC, 5d) [60,131]. 

Furthermore, antibody-based MIF-targeting strategies have been developed. The most advanced therapy, which has even reached clinical trials, is a monoclonal antibody against the oxidized form of MIF, called imalumab (Bax69, Baxter). This humanized antibody was tested in metastatic colorectal cancer patients in a phase I clinical trial (NCT01765790) and the maximum benefit was stable disease in 26% of the patients [27]. Bax69 was subsequently acquired by OncoOne and is currently being renewed into a second-generation antibody, which is being tested in pre-clinical trials. More recently, anti-MIF nanobodies (i.e., the variable region of heavy-chain-only antibodies present in the Camelidae family) have also been generated and were found to inhibit MIF’s pro-inflammatory activity. Consequently, these nanobodies were able to lower endotoxic shock-mediated lethality in a preclinical mouse model [156]. However, so far, these nanobodies have not been tested in clinically relevant tumor models. It should be noted that nanobodies are gaining more attention, especially since the first nanobody was approved for use in clinics (i.e., caplacizumab (Sanofi) for treating acquired thrombotic thrombocytopenic purpura) [157]. A few of the characteristics that make them an interesting tool are i) the lack of an Fc portion, reducing the chances of generating Fc-dependent side effects, ii) their small size, 15kDa, which allows them to bind to epitopes that are not easily accessible to monoclonal antibodies, for example, within tissues such as tumors, and iii) their high tailorability. Nanobodies can be easily formatted into bivalent (two nanobodies recognizing the same epitope), biparatopic (two nanobodies recognizing the same molecule, but different epitopes), and bispecific (two nanobodies recognizing different molecules) constructs, which may increase their potency [158,159,160]. For example, an anti-MIF nanobody has been coupled to a serum albumin-binding nanobody to increase its circulating time [156]. These characteristics confer a great advantage over antibodies and peptides, as nanobodies fall within the ideal compromise of size and side effects. 

Most blocking compounds that have been tested or that are under development are either MIF- or DDT-specific and do not allow a complete blocking of all MIF/DDT signaling. Nevertheless, there is more and more evidence from in vitro research and preclinical murine tumor models that the targeting of both MIF and DDT could have superior anti-tumor effects. In this context, a way to block all MIF/DDT/CD74 signaling would be through the direct blocking of CD74. For example, the knockdown of CD74 in gastric cancer cells significantly reduces cell proliferation [161]. CD74 deficiency was also investigated in hepatocellular carcinoma and rendered similar results, yielding reduced proliferation and a reduced number of tumors in the CD74^−/−^ mice compared with the WT controls [162]. Hence, a blockade of CD74 in both cancer cells and stromal cells appears to be beneficial. So far, only one monoclonal antibody directed against CD74, milatuzumab, has been tested in clinical trials. In 2008, it received the Orphan Drug Approval from the FDA in multiple myeloma (NCT00421525), with the results of phase I being disease stability [163]. This drug was also part of a phase I/II trial in patients with relapsed or refractory B-cell non-Hodgkin lymphoma (NHL) (NCT00868478). The treated patients had an overall modest response regarding hematological parameters and quality of life, but the major limitation of this study was the low number of patients enrolled [164]. Currently, the pharmacologic CD74 antagonist RTL1000, which competitively inhibits MIF binding and its downstream signaling, is in clinical trials for multiple sclerosis (MS) (NCT00411723). This small compound contains the extracellular domains of the MS risk factor, HLA-DR2 (DR2α1β1), linked to an autoantigenic MOG-35-55 peptide [165]. During phase I, it was determined that RTL1000 was well tolerated at a dose of ≤60 mg, and it did not worsen the MS symptoms [166].

However, it should be noted that a blockade of CD74 does not allow a complete elimination of all MIF/DDT effects, since signaling through other MIF receptors—CXCR2, CXCR4, and CXCR7—can still take place. In this context, peptides targeting both the MIF–CD74 and MIF–chemokine receptor axes have been generated and examined in vitro [149]. For instance, MIF-derived peptides able to target the CD74 ectodomain have been identified (peptide 79–86 of MIF) and tested in vitro, yet they have not been tested in disease models. The peptide C36L1, a 17-mer peptide that binds to CD74 on tumor-associated macrophages and DCs, was shown to block MIF’s immunosuppressive activities in melanoma models in vitro as well as in vivo [138]. The peptides MIF(40–49) and MIF(47–56) blocked the interaction between MIF and CXCR2, which affected the adhesion and migration of monocytes in vivo [167]. Although peptide-based therapeutics could be an alternative to small-molecule- and antibody-based strategies, they might be prone to degradation and oxidation [168]. 

## 6. Future Perspectives

We believe that cancer therapies based on targeting both MIF and DDT will have an added benefit over existing MIF-based treatments that are in advanced pre-clinical development. For example, both MIF and DDT have major immunosuppressive effects within the tumor microenvironment, which could indicate that the generation of a combinatorial treatment could be beneficial. It needs to be seen, depending on the disease state, whether the blocking of MIF and DDT should occur in a combined fashion or sequentially. In this context, the generation of compounds that either specifically block MIF or DDT, or block both simultaneously, will be important in order to assess the best in vivo strategy. In addition, it needs to be tested whether the administration of an anti-MIF/DDT blocking therapy in conjunction with other therapies focusing more on the adaptive compartment could result in a more complete boost of the immune system. 

To make rational decisions in terms of therapeutic strategy, it is of great importance to investigate more into the shared and unique roles of DDT and MIF. As stated before, MIF and DDT have similarities in structure and functionality, but it is conceivable that DDT has unique functions that still await to be discovered. Once these gaps in our understanding are filled, we may be able to delineate the exact contribution of MIF and DDT in different cancer types, as well as their exact window of operation in each condition. In turn, this could guide us toward the next steps in optimizing the blocking strategy for this therapeutically promising cytokine family.

## Figures and Tables

**Figure 1 cancers-15-00395-f001:**
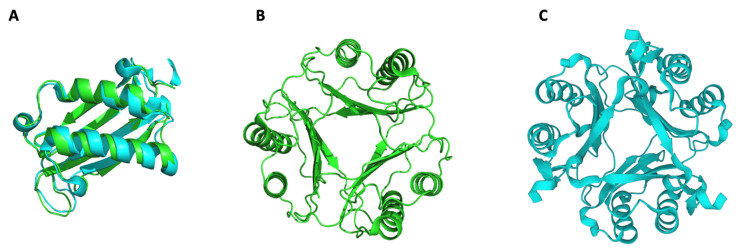
MIF and DDT structures. Reported crystal structures of human MIF and DDT. (**A**) Alignment of MIF (green) and DDT (cyan) monomers, (**B**) MIF trimer in green, and (**C**) DDT trimer in cyan. Source: RCSB-PDB. MIF: P14174, ID: 1CA7; DDT: P30046, ID: 3KAN.

**Figure 2 cancers-15-00395-f002:**
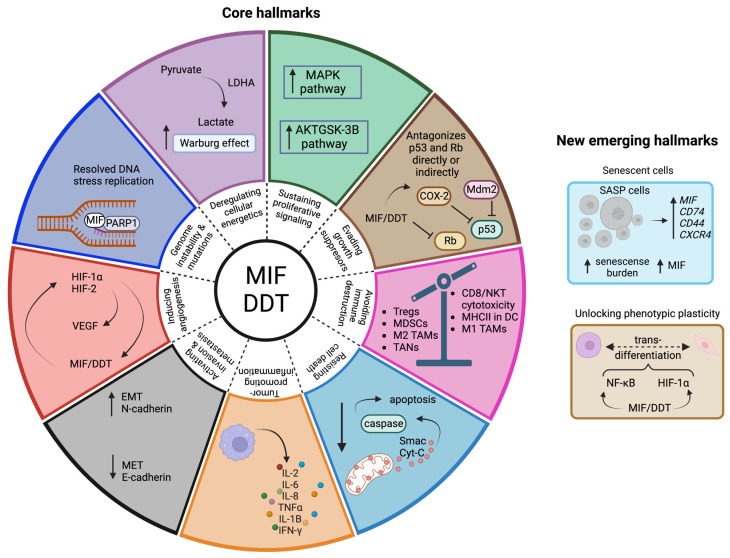
Hallmarks of cancer linked to the MIF family. The MIF family is reportedly involved in 9 out of 10 core hallmarks of cancer. Deregulating cellular energetics: MIF increases lactate production, promoting the Warburg effect. Sustaining proliferative signaling: signaling pathways involved in the expression of genes promoting proliferation are upregulated by MIF/DDT. Evading growth suppressors: p53 and Rb are antagonized directly or indirectly by MIF/DDT. Avoiding immune destruction: MIF promotes immunosuppression within the tumor microenvironment. Resisting cell death: apoptosis is inhibited by MIF/DDT. Tumor-promoting inflammation: MIF/DDT are major regulators of inflammation, promoting the expression of pro-inflammatory cytokines. Activating invasion and metastasis: MIF/DDT promote the EMT process. Inducing angiogenesis: MIF/DDT directly increase the expression of VEGF or indirectly through HIF. Genome instability and mutation: MIF acts as a 3′flap nuclease and works together with PARP in the replication fork. Concerning the more recently suggested emerging hallmarks, MIF is involved in 2 out of the 4. Unlocking phenotypic plasticity: MIF/DDT induce trans-differentiation of cells, and Senescent cells: MIF is a key regulator of senescence-associated secretory phenotype (SASP) cells. The hallmarks for which no involvement of MIF/DDT has been reported up to now are “Enabling replicative immortality”, “Non-mutational epigenetic reprogramming”, and “Polymorphic microbiomes”. Created with BioRender.

**Table 1 cancers-15-00395-t001:** Relationships of MIF and DDT with several cancer types.

Cytokine	Type of cancer	Relationship	Reference
MIF and DDT	Lung cancer—NSCLC	Both cytokines showed high levels in lung cancer patients. Promote tumor growth by increasing angiogenesis, migration, and cell proliferation and decreasing apoptosis.	[57,58,59,60]
MIF	Breast cancer	Breast cancer patients showed high expressions of MIF, especially TNBC patients, compared with non-cancer patients. MIF promotes tumor growth by preventing apoptosis and increasing cell proliferation and metastasis.	[61,62]
MIF	Prostate cancer	Serum MIF levels were significantly higher in prostate cancer patients compared with no-prostate cancer patients. MIF promotes tumor growth by inducing cancer cell growth, invasion, and angiogenesis.	[63,64,65]
MIF and DDT	Cervical cancer	DDT showed significantly increased levels in cervical cancer patients; these levels were correlated with lymph node metastasis. MIF was found in high levels in cervical intraepithelial neoplasia and even higher levels in cervical squamous cell carcinomas. Both cytokines promote tumor growth by modulating proliferation, migration, and cell invasion.	[12,66]
MIF	Hepatocellular carcinoma	MIF expression was highly present in tumor samples compared with non-tumor tissues. High levels of MIF were associated with a poorer prognosis. MIF might be involved in tumor progression, migration, invasion, angiogenesis, and metastasis.	[67,68,69]
MIF	Gastric cancer	MIF levels were higher in cancer tissues compared with adjacent non-cancer tissues. High levels were correlated with advanced stages, lymph node metastasis, and poor survival. MIF promotes tumor growth by supporting cancer cell proliferation.	[70,71,72]
MIF	Esophageal cancer	MIF levels were increased in tumor samples compared with non-tumor samples.	[73]
MIF and DDT	Glioblastoma	High levels of MIF and DDT were associated with reduced patient survival and poor prognosis. Blocking MIF reduced myeloid-derived suppressor cell (MDSC) activity, decreased Treg numbers, and increased CD8+ T cell function. MIF inhibition promoted IFN-γ release, leading to tumor growth inhibition and a glioma-associated microglia polarization from an M2 to M1 phenotype. An all-MIF blocking strategy increased the radiation therapeutic effect.	[11,74,75,76]
MIF and DDT	Renal carcinoma	MIF and DDT showed significant levels in renal cancer patients. Both cytokines promote tumor growth by increasing cell survival, growth, and migration.	[77]
MIF and DDT	Pancreatic ductal adenocarcinoma	MIF and DDT were found at significant levels in cancer tissues compared with normal tissues. The expressions of both cytokines showed a significant correlation. Both cytokines are involved in tumor progression by inhibiting apoptosis and increasing cell proliferation and invasion.	[78]
MIF and DDT	Neuroblastoma	High levels of MIF and DDT in samples of stage IV patients were associated with poor prognoses. MIF promotes tumorigenesis likely by decreasing antigen presentation and cytotoxic responses.	[79]
MIF and DDT	Melanoma	Tumor-bearing mice had higher DDT levels compared with naïve mice. High levels of MIF in metastatic melanoma lesions correlated with faster disease progression. MIF and DDT increase tumor growth by modulating proliferation and apoptosis.	[80,81]
MIF and DDT	Ovarian cancer	High levels of DDT were found in ovarian cancer compared with the control group; these levels showed a strong correlation with MIF levels. Levels of MIF correlated with tumor stage and platinum sensitivity and the infiltration of CD8+ T- and NK-cells into the tumor.	[41,82]

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
