# Peer review of "Hallmarks of Cancer Affected by the MIF Cytokine Family"

_cancers, 2023, doi:10.3390/cancers15020395_

Round 1
Reviewer 1 Report
see attached file

Author Response
Replies to the reviewers’ comments:
Reviewer 1
Overall, this is very well-rationalized and well-written manuscript, and I am grateful for the opportunity to review and learn from it. The manuscript will likely form a fundamental reference for the field going forward, and influence in a positive way research direction in the field.
We thank the reviewer for the very supporting comment.
1) Lines 83- 104. The authors are cautioned to temper markedly their synthesis of the “oxMIF” phenomena. First, the molecular identity of the putative oxidized epitope remains to be demonstrated in vivo or in vitro, and the implications of MIF oxidation are inferred and highly speculative. (Indeed, the authors do not mention the description of a myeloperoxidase produced, Pro1 oxidation of MIF, for which there is at least molecular evidence published). Second, the published evidence in support of oxMIF is immunochemical, and whether “oxidation” is simply a surrogate for an induced or conformational epitope is unknown (but suspected given the high conformational dynamics of the MIF structure). Third, the little published data available suggests that oxMIF is less bioactive than native MIF. Accordingly, how to rationalize a pathogenic or cancer-specific role for oxMIF?
We thoroughly altered the section on oxMIF, based on the reviewer’s concerns. Main adaptations are:
Line 93-94: addition of following sentence: “Of note, the implications of MIF oxidation are still debated in the field.”
Line 96-103: inclusion of the data regarding the MPO-mediated Pro1 oxidation + reference 24
Line 106-110. “Remarkably, it has been suggested that redMIF could represent the latent form of MIF, whereas oxMIF might be the biologically active isoform [22]. However, given that Pro-1 is essential for CD74 binding, it could be argued that oxMIF (or at least MIF oxidized at Pro-1) is potentially less potent in triggering CD74 signaling, thereby annihilating several of MIF’s functions.” This addition somewhat contrasts the statement that oxMIF would be the biologically active form, while not being able to bind to CD74.
Line 111-112: following statement added: “Though the molecular identity of the putatively oxidized epitope remains to be demonstrated, certainly in vivo”.
Line 121: following statement added: “oxMIF has been proposed as the “bad MIF”, which is however speculative”
Line 126-127: following statement added: “However, the exact contribution of oxMIF in cancer, if any, as well as in inflammation remains to be determined.”
2) Line 133. The human cell based report for MIF activation of NLRP3 should be cited (Shin et al, 2019 A&R).
This paper has now been cited as reference 36 on line 158
3) Line 138. “On top of that” (not in written English). Please delete.
Has been deleted.
4) Line 173. Remove the teleologic phrasing; “the body needs . . .” This should be removed. Suggested replace with: MIF-specific regulatory mechanisms likely exist.
Has been replaced.
5) Line 180. Delete “will”; replace “affect “ with “affects”.
Has been changed.
6) Line 195. Include recent biologic studies regarding MIF ELR by Tilstam et al. JCI 2021.
The work by Tilstam et al (JCI 2021) has now been described in lines 219-225. The paper has been included in the reference list, as reference 53.
7) Line 271. Section 4.1. Key support for MIF/p53 is genetic, provided by MIF-KO mouse studies and the initial report by Fingerle et al. PNAS 2003.
The work by Fingerle has now been described in the manuscript (lines 292-296) and has been added to the reference list (reference 87).
8) Line 467. First report of DDT specificity is ref 135, which should be included here.
This reference (now ref 131) has been included at this position in the manuscript (lines 510-513).
9) Line 469. “For long,” (for long what?). Please re-phrase.
Has been rephrased into “Inflammation and cancer have an intricate relationship, but more information has been gathered in the last decade to understand better how inflammation impacts tumorigenesis” (lines 520 -522)
10) Line 489. To state that MIF is a major immunosuppressive factor is incorrect. It is immunosuppressive in certain contexts, possibly in cancer. Here I suggest citing the comprehensive studies of Alban & Lathia.
We now specified that the immunosuppressive role of MIF may be context-dependent:
Lines 539-540: “Besides being an initiator of inflammation, MIF is also reported as a major immunosuppressive factor [103], although this may be restricted to certain contexts such as cancer.”
We included a reference by Alban&Lathia as reference 74.
11) Line 539. Delete the redundant phrase: “in any living organism.”
Has been deleted.
12) Lines 546 – 547. The size of a small molecule pharmacophore is irrelevant to clearance rates or dosing, which is governed by metabolism and elimination. Please re-phrase/correct.
This section has been modified. Lines 653-656: “Due to the fact that these small compounds are non-immunogenic, have a low manufacturing cost, efficient tissue penetration, and can be given orally, this pipeline of inhibitors has gained great interest. However, they show a different half-life compared to monoclonal antibodies due to differences in their elimination rate and metabolic conversion.”
13) Line 552. “As a matter of fact. . “ (what fact?) Please delete this phrase.
This phrase has been deleted.
14) Page 15. Mention of the CD74 antagonist RTL1000, which has completed phase I testing (albeit in MS not cancer) should be made, with citation.
Findings on RTL1000 have been added to the manuscript (Lines 726-731) and as references 165 and 166.
15) Line 596. Please update information about Milatuzumab. It is now US FDA approved for multiple myeloma (and has been studied in lupus nephritis).
The updated information on Milatuzumab has been included in Lines 719-726 and as references 163 and 164.
Reviewer 2 Report
In this manuscript (cancers-2089178), Barthelmess, Stijlemans, and Van Ginderachter provide an overview of the macrophage migration inhibitor factor (MIF) family in cancer (MIF and DDT/MIF2), organized around the classical hallmarks of cancer. This review is timely and there appears to be a need for such a review for the field as there has not been a recent general review on MIF and cancer. Overall, this review is well-written and informative. Below are some suggestions to consider that will increase the impact of this review and provide some additional clarity.
1. The concept of the paper is organized around the hallmarks of cancer, which is an interesting angle, but it would be useful if the review was updated around the newest version (PMID 35022204). When this is updated, it may also be worth including all hallmarks as there is some evidence, albeit it weak, that MIF signaling is important for colon cancer cell proliferation (PMID 35567291) and directly binds to p53 in glioblastoma (PMID 26980763).
2. The summary of the tumors where MIF has been studies (Table 1) is generally good but missing glioblastoma. There are multiple studies in this tumor type and it would be worth including, including but not limited to PMIDs 34516577, 32625208, 30814573, 27157615, 27145382, 26980763
3. In the discussion about MIF inhibition strategies, Ibudilast should also be mentioned (p. 15). It is currently under assessment for glioblastoma (NCT03782415).
4. In the simple summary, the concept that the review will “elucidate” the available information is confusing, it may be best to simply delete “elucidate and.”
5. There appears to be an aberrant green highlight on line 560.
Author Response
This review is timely and there appears to be a need for such a review for the field as there has not been a recent general review on MIF and cancer. Overall, this review is well-written and informative.
We thank the reviewer for the positive comments.
- The concept of the paper is organized around the hallmarks of cancer, which is an interesting angle, but it would be useful if the review was updated around the newest version (PMID 35022204). When this is updated, it may also be worth including all hallmarks as there is some evidence, albeit it weak, that MIF signaling is important for colon cancer cell proliferation (PMID 35567291) and directly binds to p53 in glioblastoma (PMID 26980763).
We now describe and refer to the newest version of the hallmarks of cancer (reference 84). Two of these suggested new hallmarks could also be linked to MIF. Consequently, we adapted Figure 2 (including the suggested hallmarks “Unlocking phenotypic plasticity” and “Senescent cells”) and wrote two entirely new sections (sections 4.10 and 4.11) on the possible involvement of MIF/DDT in these novel hallmarks.
- The summary of the tumors where MIF has been studies (Table 1) is generally good but missing glioblastoma. There are multiple studies in this tumor type and it would be worth including, including but not limited to PMIDs 34516577, 32625208, 30814573, 27157615, 27145382, 26980763
We included glioblastoma in Table 1, and the suggested references can be found in the reference list as numbers 122, 11, 74, 154, 75, 76, and 88.
- In the discussion about MIF inhibition strategies, Ibudilast should also be mentioned (p. 15). It is currently under assessment for glioblastoma (NCT03782415).
The work performed with Ibudilast has now been included at several places in the manuscript:
Lines 579-581, describing the use of Ibudilast to reduced MDSC functionality in glioblastoma (reference 74)
Lines 673-680, describing its approval in Japan to treat bronchial asthma and cerebrovascular disorders; and its use in a clinical trial to treat glioblastoma, in combination with temozolomide.
- In the simple summary,the concept that the review will “elucidate” the available information is confusing, it may be best to simply delete “elucidate and.”
“elucidate and” has been deleted
- There appears to be an aberrant green highlight on line 560.
This has been removed.